# Generating High-Power, Frequency Tunable Coherent THz Pulse in an X-ray Free-Electron Laser for THz Pump and X-ray Probe Experiments

Yin Kang [1,2], Zhen Wang [3], Kaiqing Zhang [3,*] and Chao Feng [1,3]

1   University of Chinese Academy of Sciences, Beijing 100049, China
2   Accelerator Physics and Laser Technology Department, Shanghai Institute of Applied Physics, Chinese Academy of Sciences, Shanghai 201800, China
3   Accelerator Physics and Laser Technology Department, Shanghai Advanced Research Institution, Chinese Academy of Sciences, Shanghai 201210, China
*   Correspondence: zhangkaiqing@zjlab.org.cn

**Abstract:** Precisely synchronized X-ray and strong-field coherent terahertz (THz) enable the coherent THz excitation of many fundamental modes (THz pump) and the capturing of X-ray dynamic images of matter (X-ray probe), while the generation of such a light source is still a challenge for most existing techniques. In this paper, a novel X-ray free-electron laser based light source is proposed to produce a synchronized high-powered X-ray pulse and strong field, widely frequency tunable coherent THz pulse simultaneously. The technique adopts a frequency beating laser modulated electron bunch with a Giga-electron-volt beam energy to generate an X-ray pulse and a THz pulse sequentially by passing two individual undulator sections with different magnetic periods. Theoretical analysis and numerical simulations are carried out using the beam parameters of the Shanghai soft X-ray free-electron laser facility. The results show that the technique can generate synchronized 4 nm X-ray radiation with a peak power of 1.89 GW, and narrow-band THz radiation with a pulse energy of 1.62 mJ, and the frequency of THz radiation can be continuously tuned from 0.1 to 40 THz. The proposed technique can be used for THz pump and X-ray probe experiments for dynamic research on the interaction between THz pulse and matter at a femtosecond time scale.

**Keywords:** coherent THz radiation; X-ray radiation; free-electron laser (FEL); frequency beating





## 1. Introduction

Terahertz (THz) radiation, with a frequency from 0.1 to tens of THz, has excellent applications in the fields of information and communication technology, high-resolution sensing, THz astronomy, and non-destructive evaluation, among others [1–5]. More importantly, it can serve as a unique tool for exploring many fascinating scientific frontiers, due to its extraordinary frequency and timescale properties [6,7]. THz radiation can provide direct access to study the fundamental modes, including motions of free electrons, the rotations to molecules, the vibrations of crystal lattices, superconducting oscillation and the precessions of spins, by interacting with various freedom degrees of matter and thus providing a versatile handle for the control of matter. Recently, the rapid development of strong-filed THz sources with a field strength on the order of MV/cm has opened a new era for understanding a variety of scientific phenomena, such as strong field-matters interactions, the high-harmonic generation of THz waves, nonlinear THz spectroscopy, control of THz wave-induced fluorescence for remote sensing, and giant non-linearity in the THz frequency range [8–10]. After matter to be controlled is excited by a THz pulse, a time-delayed femtosecond (fs) to picosecond (ps) pulse is generally required to measure its instantaneous state and observe the dynamic process induced by the THz pulse. Combining with an X-ray pulse, one can capture the dynamic image of matters by X-ray scattering or

diffraction, which is the so-called THz pump and X-ray probe technique. The technique has been extensively employed in several aspects: on the one hand, the technique can measure the basic properties of matters such as magnetization, conductivity and even crystal lattice by adjusting the time delay between THz and X-ray pulse [11–13]. On the other hand, the technique can also detect the duration and time structure of an individual X-ray pulse by using a THz-field-driven X-ray streak camera [14] and measuring the dynamic process of matter by performing time-resolved X-ray diffraction [15].

The prominent issue holding back the development of THz pump and X-ray probe experiment is the lack of synchronized strong-field THz-pump and X-ray-probe sources. Until now, strong field THz radiation has been mainly produced by ultrafast laser, laser-produced plasmas and electron accelerator-based techniques [16–25]. However, the pulse energy is limited to several hundred microjoules (μJ) for the ultrafast laser technique [16,17]. The laser-produced plasmas technique can produce THz radiation at a millijoule (mJ) level, while the spectral coverage is generally quite limited [18–21]. Most importantly, neither of these two techniques can generate X-ray pulses and THz pulses simultaneously. The electron accelerator can produce THz pulses by several ways: coherent synchrotron radiation (CSR) [26–28], coherent transition radiation (CTR) [29–31], or undulator radiation [32–34]. However, the maximum pulse energy of THz radiation is several hundred μJ, and the pulses normally have a wide spectral bandwidth for CSR and CTR techniques. Undulator radiation, such as free-electron laser (FEL) [35–38], has been recognized as a reliable technique that holds the capability of generating ultra-short pulses with gigawatt (GW) level peak power and a tunable wavelength from THz to X-ray. A variety of undulator-based mechanisms have been proposed to generate intense THz pulses, although most of them cannot be used to produce synchronized THz pulse and X-ray pulse at the same time. Now, THz radiation in a high gain X-ray FEL facility is one of the main choices for generating synchronized THz and X-ray pulses. FEL facilities around the world, including LCLS-II [39,40], FLASH [41–43], European-XFEL [33,44,45] and Swiss-FEL [46–48] have adopted an afterburner to compress the duration of the electron beam into one THz period to obtain high-power X-ray and intense THz with a broad spectral bandwidth of about 10%, a pulse energy up to 200 μJ and a great synchronization performance. However, generating THz radiation with a frequency above 10 THz requires the suppression of the pulse duration of the electron beam below 100 fs, which is still a challenge for most of the existing FEL facilities.

In this paper, a novel technique is proposed to generate a synchronized high-power X-ray pulse and narrow-band THz pulse with a pulse energy of millijoule (mJ) level and tunable frequency at the full THz range. Previously, we proposed a compact accelerator based light source with an electron beam of several tens of Mega-electron-volts (MeV) for high-power, full bandwidth tunable coherent THz generation, which combines the techniques of frequency beating laser modulated electron beam and undulator radiation [49]. Here, we adopt a frequency beating laser modulated electron bunch with a Giga-electron-volt (GeV) level beam energy to generate an X-ray pulse and a THz pulse sequentially by passing two individual undulators with different magnetic periods. In this proposed technique, the X-ray pulse and THz pulse are generated using the same electron bunch, and thus the two pulses can have intrinsic synchronization performance. Inheriting the basic performance of X-ray FEL facility, the proposed can generate a GW level X-ray pulse. Due to the enhanced peak current, the modulated electron bunch can improve spontaneous amplified self-emission (SASE) radiation performance. At the same time, the proposed technique can produce coherent THz radiation with tunable frequency from 0.1 to 40 THz, and pulse energy of several mJ, by adjusting the laser beating frequency and the THz undulator gap. Using the strong-field THz to excite matters and measure its instantaneous state, THz pump and X-ray probe experiments can be performed to explore various scientific frontiers. This paper is organized as follows: the principles of the proposed technique with high order dispersion effects are introduced in Section 2. Using typical parameters of Shanghai soft X-ray free-electron laser facility (SXFEL), start-to-end simulation results are presented in Section 3. Finally, some concluding comments are given in Section 4.

## 2. Working Principle

The layout of the frequency beating laser system in the proposed technique is shown in Figure 1. To illustrate the process of frequency beating, a transform-limited Gaussian pulse with center angular frequency at $\omega_0$, amplitude $A$ and pulse width $\sigma_0$ is considered, and the electric field is given by

$$E_{in}(t) = A \exp(\frac{-t^2}{\sigma_0^2} + iw_0 t). \tag{1}$$

In the optical system, an ultrafast laser is stretched by a parallel grating pair, and dispersion effects will be introduced at the same time. These dispersion effects are the phase modulations with Fourier components, which can be written as Taylor series expansions at the center angular frequency $\omega_0$ [49–51]

$$\phi(w) = \phi(w_0) + \phi_1(w - w_0) + \phi_2 \frac{(w - w_0)^2}{2!} + \phi_3 \frac{(w - w_0)^3}{3!} + \cdots . \tag{2}$$

Here, $\phi_1$ is the group delay at $\omega_0$, $\phi_2$ is the second order dispersion or the group delay dispersion (GDD), and $\phi_3$ is the third order dispersion (TOD). Among them, the GDD provides a linear chirp, which plays leading roles in the processes of broadening and compressing the laser pulse. High order dispersions such as the TOD mainly cause distortion of the pulse longitudinal structure. In this analysis, dispersions above the third order are ignored, since it is relatively smaller compared with the TOD [50,52]. After passing through the grating pair, the phase modulated output pulse is given by [49–51]

$$E_{out}(t) = \frac{1}{2\pi} \int_{-\infty}^{\infty} \exp(-iwt) \exp[i\phi(w)] dw \times \int_{-\infty}^{\infty} \exp(iwt') E_{in}(t) dt'. \tag{3}$$

When substituting Equations (1) and (2) into Equation (3) and only considering the GDD and TOD terms, the electric field of the output laser pulse can be obtained by

$$E_{out}(t) = \frac{A}{2} (\frac{\sigma_0}{\sigma_{out}})^{\frac{1}{2}} \exp(-\frac{t^2}{\sigma_0^2}) \exp[i(w_0 t + \alpha t^2 + \beta t^3)]. \tag{4}$$

Here, the broadened pulse width $\sigma_{out} = \sigma_0 (1 + \phi_2^2/\sigma_0^4)^{\frac{1}{2}}$ and the chirp parameter $|\alpha| = 1/\sigma_0 \sigma_{out}$, and the cubic phase parameter $\beta = \phi_3/6\phi_2^3$.

According to Figure 1, the GDD is given by

$$\phi_2 = -\frac{4\pi^2 cL}{w_0^3 d^2} [1 - (\frac{2\pi c}{w_0 d} - \sin\theta)^2]^{-\frac{3}{2}}, \tag{5}$$

where $L$ is the distance between the grating interfaces and $d$ is the grating constant, and $\theta$ is the incident angle. The TOD is given by

$$\phi_3 = -3\frac{\phi_2}{w_0} [\frac{1 + \frac{2\pi c}{w_0 d} \sin\theta - \sin^2\theta}{1 - (\frac{2\pi c}{w_0 d} - \sin\theta)^2}]. \tag{6}$$

After passing through the grating pair, the stretched laser pulse is divided into two branches by an optical splitter. Then, the time delay $\tau$ between the two pulses is introduced by a tunable optical delay line. Finally, the two pulses are recombined by an optical mixer to obtain a frequency beating laser pulse, which contains Fourier components with THz frequency. The intensity of the recombined pulse is given by

$$I_{total} = \frac{1}{2} \left| E_{out}(t + \frac{\tau}{2}) + E_{out}(t - \frac{\tau}{2}) \right|^2 = I_+(t) + I_-(t) + I_{cross}(t). \tag{7}$$

Here, $I_{\pm}(t)$ are the unessential low-frequency (DC) components in the Fourier frequency domain and appear as a slowly varying Gaussian envelope in the time domain. The cross term $I_{cross}(t) \propto \cos(w_0 t + \alpha t^2 + \beta t^3)$ represents the quasi-sinusoidal chirp modulation of the light intensity at the beating frequency $f$, which is given by [50]:

$$f = \frac{\tau}{\pi \sigma_0 \sigma_{out}} \cong \frac{|\alpha|\tau}{\pi}. \tag{8}$$

According to Equation (8), the beating frequency $f$ can be continuously tuned by adjusting the chirp parameter $|\alpha|$ and the time delay $\tau$.

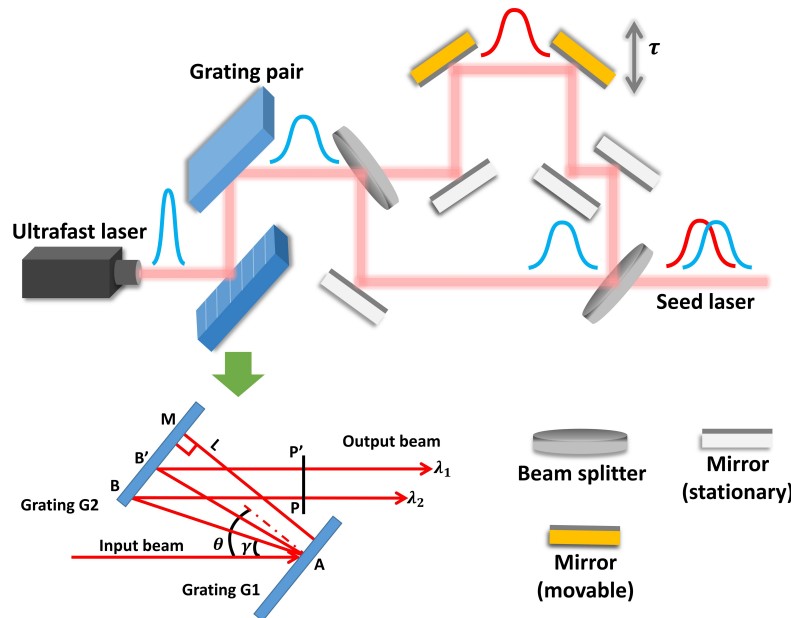

**Figure 1.** The layout of the frequency beating laser system in the proposed technique.

The schematic layout of the proposed technique is shown in Figure 2. In the proposed technique, an electron beam with an average beam energy of $E_0$ and dimensionless energy deviation $P = (E - E_0)/\sigma_\gamma$ is produced from a LINAC accelerator; here, $\sigma_\gamma$ is the energy spread and $E$ is the beam energy. The electron beam and the frequency beating laser pulse are firstly sent into a modulator to interact with each other and obtain the energy modulation at THz frequency. The dimensionless energy deviation $P_1$ becomes

$$P_1 = P + A_0 \sin \frac{wZ}{c}, \tag{9}$$

where $w$ is the frequency of the seed laser, $c$ is the speed of light in a vacuum, $Z$ is the longitudinal position along the beam, and $A_0$ is the energy modulation amplitude normalized to the energy spread. Then, the electron beam passes through a magnetic chicane with a dispersion strength $R_{56}$ to convert the energy modulation into density modulation, and the longitudinal position $Z_1$ becomes

$$Z_1 = Z + R_{56} P_1 \frac{\sigma_\gamma}{E_0}. \tag{10}$$

After that, the modulated electron beam is sent into an undulator with a relatively short period to produce X-ray radiation until saturation. The density modulation will not decrease the amplification of the X-ray radiation, due to the three orders of magnitude difference in wavelength between THz and X-ray. Finally, the electron beam passes through a wiggler with a relatively long period to produce high-power, coherent THz radiation. The

electron beam can be used to generate THz radiation, since the required energy spread for THz radiation is much larger than that of X-ray. By adjusting the chirp parameter $|\alpha|$ from 3.13 ps$^{-2}$ to 78.3 ps$^{-2}$ and the time delay $\tau$ from 0.1 ps to 0.8 ps, the THz wavelength can be continuously tuned from 0.1 to 20 THz. Considering the actual Wiggler designer at SXFEL, the wiggler is only used to obtain THz radiation from 5 to 20 THz for the fundamental radiation, and the THz radiation from 0.1 to 5 THz is generated by CTR. In addition, the frequency range of THz radiation can be easily extended to 40 THz by using a harmonic lasing technique [53].

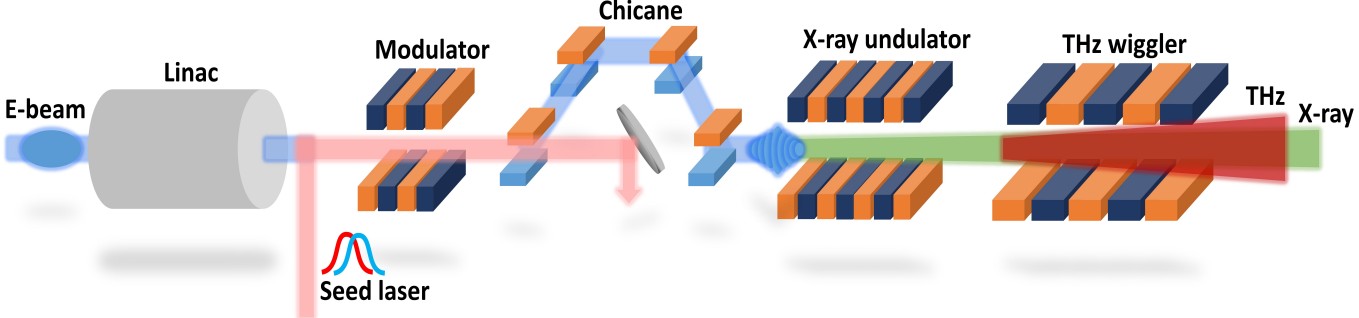

**Figure 2.** The layout of the proposed technique.

## 3. Results

### 3.1. Generation of Frequency Beating Laser and Laser-Beam Modulation

To illustrate the feasibility of the proposed technique, simulations are performed with parameters of laser system and electron beam in Table 1 using the code of ASTRA [54], FALCON [55], ELEGANT [56] and GENESIS 1.3 [57]. The laser frequency beating and the laser-beam modulation process are simulated using a laser pulse with the central wavelength of 800 nm and an initial pulse duration of 15 fs. In the simulation, the laser pulse is stretched to 0.85 ps by a grating pair and a time delay of 0.8 ps is introduced by an optical time delay line. At the end of the laser system, the longitudinal amplitude distribution and spectrum of the frequency beating laser pulse are shown in Figure 3. One can observe that the optical signal at the THz frequency in the longitudinal distribution is produced and the fundamental wavelength of frequency beating is about 15 μm, which fits well with the calculated results from Equation (8). The frequency beating laser pulse can be treated as a seed laser to modulate the electron beam.

**Table 1.** The parameters of the laser system and electron beam.

| Parameters | Value |
| --- | --- |
| Laser wavelength | 800 nm |
| Laser pulse width $\sigma_0$ | 15 fs |
| Peak power | 9 MW |
| Grating line | 250 mm$^{-1}$ |
| Incident angle | 35° |
| Grating pair distance | 9 cm |
| Electron beam energy | 1.4 GeV |
| Energy spread (slice) | 0.001% |
| Bunch length | 570 μm |
| Bunch charge | 500 pC |
| Undulator period | 1.6 cm |
| Wiggler period | 20 cm |

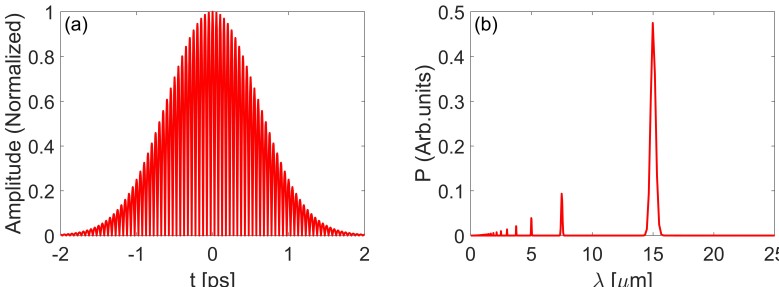

**Figure 3.** The longitudinal amplitude distribution (**a**) and beating frequency (**b**) of the seed laser.

Start to end (S2E) simulations are carried out to present the possible performance of the proposed technique with the parameters of SXFEL. Firstly, the generation and dynamics of the electron beam in the photoinjector are simulated with the code of ASTRA, where a longitudinal uniform electron beam can be obtained with a charge of 500 pC and a pulse length of about 8 ps. Then, the electron beam is compressed to about 350 A and boosted to about 1.4 GeV by a magnetic compressor and the accelerating structures in the LINAC. The simulations are performed with the code of ELEGANT, and the longitudinal phase space and the current profile of the electron beam at the exit of the LINAC are shown in Figure 4, where one can find that there is a quasi-linear energy chirp and Gaussian-like current profile along the longitudinal position of the electron beam.

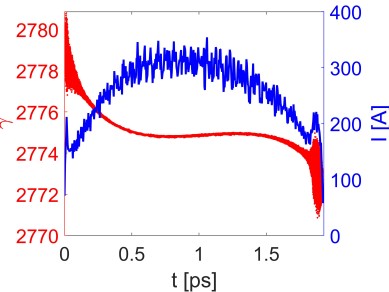

**Figure 4.** The longitudinal phase space ($\gamma$ red line) and the current profile ($I$ blue line) of the electron beam.

After that, the electron beam together with the frequency beating laser are sent into the modulator with a magnetic period of 10 cm to interact with each other and obtain an energy modulation at the same wavelength with the frequency beating laser (about 15 μm). Then, the modulated electron beam passes through the magnetic dispersion section with an $R_{56}$ of 8.18 cm to convert the energy modulation into density modulation. The simulations are carried out using the code of FALCON, and the longitudinal phase spaces of the electron beam before and after the dispersion section are presented in Figure 5. To show the details of energy and density modulation clearly, only parts of the longitudinal phase spaces are presented. In addition, the corresponding current profile and the corresponding bunching factor are also given. From Figure 5, the envelope of the electron beam before the dispersion section is energy modulated, and after the dispersion section the energy modulation is effectively converted into density modulation. In addition, one can observe that the longitudinal distribution of the electron beam has an apparent bunching at 15 μm and its harmonics, and the bunching factors at 15 μm (20 THz) and 7.5 μm (40 THz) are about 0.30 and 0.15, respectively. At the same time, the peak current of the electron beam has been enhanced from 354 A to 542 A due to the density modulation, which can shorten the gain length and increase the final saturation power of the X-ray radiation.

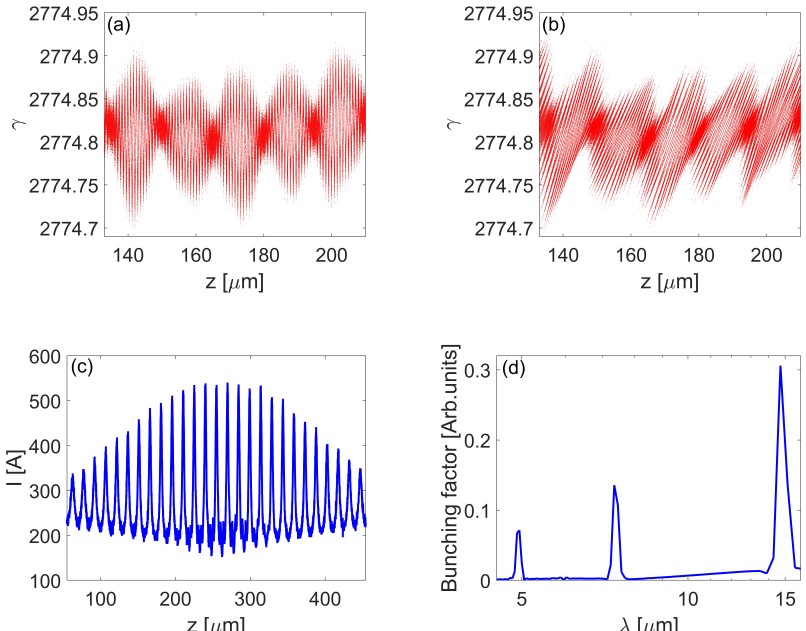

**Figure 5.** The longitudinal phase spaces before (**a**) and after (**b**) the dispersion section. The current profile (**c**) and the bunching factor (**d**) after the dispersion section.

### 3.2. Generation of X-ray and THz Radiation

The electron beam is then sent into undulators with magnetic periods of 1.6 cm and 20 cm to produce X-ray and THz radiation sequentially. The electron beam first passes through an undulator with a magnetic period of 1.6 cm and a dimensionless undulator parameter $K$ of 2.38 to generate X-ray radiation at 4 nm, and the process is simulated with the code of GENESIS 1.3. Figure 6 shows the X-ray peak power evolutions in the undulator for the conventional SASE. From Figure 6, one can find that the saturated X-ray peak power is 1.89 GW at the undulator position of 19 m with the proposed technique, and 1.45 GW at the undulator position of 36 m for conventional SASE, which indicates that the gain length is significantly reduced and the saturated peak power is enhanced with the proposed technique, due to the enhanced peak current.

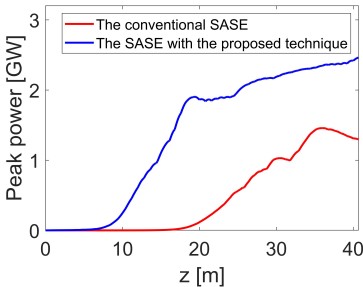

**Figure 6.** The X-ray peak power evolutions of the conventional SASE radiation and SASE radiation with the proposed technique.

After that, the electron beam is sent into a wiggler with a magnetic period of 20 cm and a dimensionless undulator parameter $K$ of 48.03 to generate THz radiation at 15 μm. The electron beam at the exit of the X-ray undulator can still be used to generate THz radiation due to the energy spread requirement difference between X-ray and THz radiation. In addition, the electron beam can also be used to produce THz radiation at 7.5 μm by adjusting the gap of the wiggler, since the bunching of the electron beam contains Fourier components at second harmonics according to Figure 5. Figure 7 shows the peak power

evolutions, bunching factor evolutions and the spectra of THz radiation at 15 μm and 7.5 μm, where the peak power of 15 μm radiation can become saturated at 15 m with a peak power of 2.41 GW and a bunching factor of 0.75. According to Figure 7c, the FWHM of the saturated spectral bandwidth is about 0.49 μm and the maximum pulse energy is calculated to be 1.62 mJ. For the 7.5 μm radiation, the peak power can become saturated at 18 m with a peak power of 1.98 GW and a bunching factor of 0.74. Correspondingly, the FWHM of spectral bandwidth is 0.22 μm and the maximum pulse energy is 1.49 mJ.

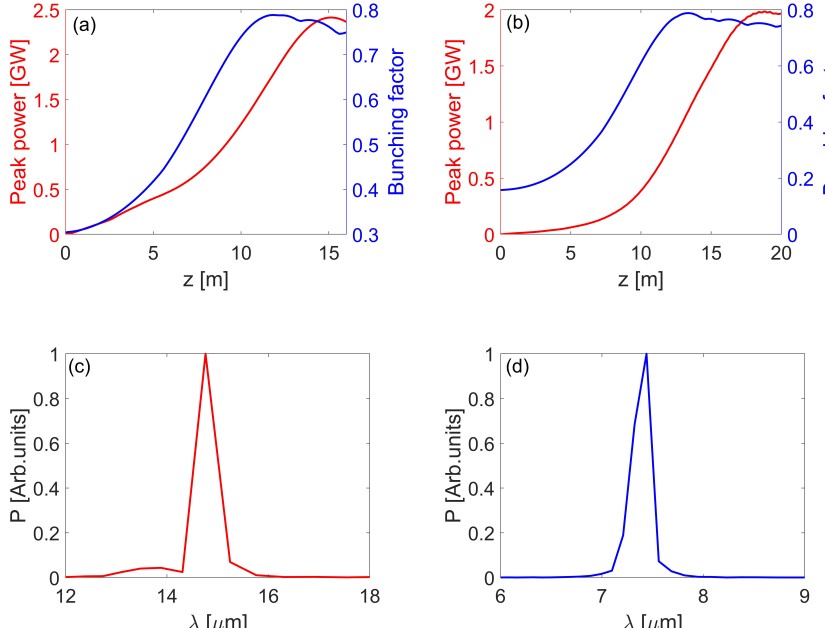

**Figure 7.** The evolutions of peak power and bunching factor at 15 μm (**a**) and 7.5 μm (**b**). The THz radiation spectra at 15 μm (**c**) and 7.5 μm (**d**).

In this proposed technique, the laser pulse is stretched by a grating pair, which will inevitably introduce high order dispersion effects. The influence of high order dispersions on a beat frequency-based THz light source has been analyzed in our previous work [58]. The results indicate that the high order dispersion effects have little influence on the generation of THz radiation. Here, the influence of high order dispersions on the proposed technique is also analyzed. The analysis adopts an experimental measured TOD of $2.40 \times 10^{-6}$ ps$^3$ and GDD of $-6.38 \times 10^{-3}$ ps$^2$, which uses the technique of state-of-the-art laser control [59]. Figure 8 shows the amplitude distribution and the spectrum of the beating profile with the TOD. By comparing Figures 3 and 8, the high order dispersion effects mainly lead to nonlinear chirp on the envelope of the frequency beating laser, and the spectra for all harmonics have been broadened and undergone slight wavelength shifts.

However, because the X-ray radiation comes from the initial noise inside the electron beam, the high order dispersion effects have little influence on the X-ray radiation. The THz radiation signal originates from the envelope modulation of the frequency beating laser pulse to the electron beam. When the structures of the frequency beating light pulse are mapped to the electron beam, the electron beam appears with heterogeneous bunching structures at the beating frequency. These non-uniform bunching structures will lead to the broadening and slight wavelength shift of the THz radiation spectrum. In Figure 9, the results of THz radiation with or without the TOD are shown.

According to Figure 9, one can find that the impacts of the high order dispersion effects are small. The peak power of THz radiation decreases slightly from 2.41 GW to 2.23 GW when the TOD is considered, and the FWHM spectral bandwidth of THz radiation broadens from 0.49 μm to 0.89 μm. In addition, the spectrum has a slight wavelength shift

from 15 μm to 14.54 μm and the maximum pulse energy decreases from 1.62 mJ to 1.52 mJ. In the case of high order dispersion compensation, the pulse energy of narrow-band THz radiation can still remain at the mJ level.

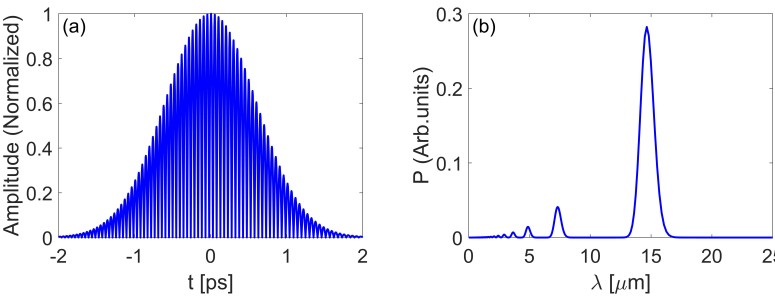

**Figure 8.** The amplitude distribution (**a**) and spectrum (**b**) of the frequency beating profile with the TOD.

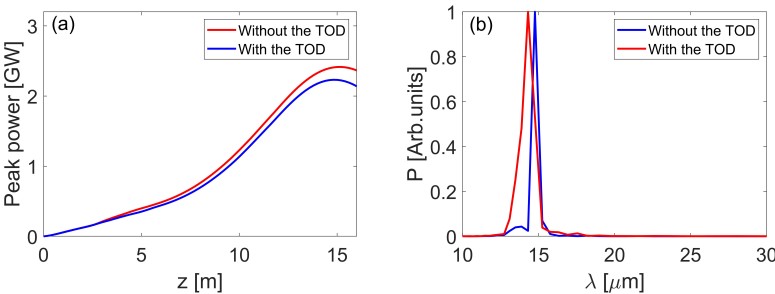

**Figure 9.** The evolutions of peak power (**a**) and spectra (**b**) of the THz radiation with or without the TOD.

## 4. Conclusions

In conclusion, we propose an X-ray FEL-based technique using the single electron beam to generate X-ray and THz radiation simultaneously, and the influence of high order dispersion effects on this technique is analyzed. Compared with our previous work on compact strong field THz sources [49], this proposed technique can produce 4 nm X-ray radiation with peak power up to 1.89 GW, and the saturated peak power is higher than that of conventional SASE radiation due to the enhancement of the beam peak current by THz modulation. At the same time, the proposed technique can also generate coherent THz radiation at 15 μm (20 THz) with a peak power up to 2.41 GW and a maximum pulse energy of 1.62 mJ. By combining harmonic lasing techniques, it can also obtain THz radiation at 7.5 μm (40 THz) with a maximum pulse energy of 1.49 mJ. The frequency of THz radiation for the proposed technique can be tuned from 0.1 to 40 THz by adjusting the laser beating frequency. Since a grating pair is used to stretch the light pulse, high order dispersion effects are inevitably introduced. When the high order dispersions are compensated, the influence of high order dispersions on THz radiation is limited and the pulse energy of coherent THz radiation remains at mJ level.

In the simulations above, an electron beam with a charge of 500 pC is used. The final pulse energy of THz radiation can be further increased by properly increasing the charge of the electron beam. As the strong field THz pulse and X-ray pulse are generated from the same electron beam, they are naturally synchronized on the fs scale, and the opening angle of the generated THz radiation is considerably larger than that of the X-ray [60]. The delay of the X-ray radiation can be achieved by a reliable optical path design [60,61], so that the proposed technique can satisfy the requirements of THz pump X-ray probe experiments, which can be used to study the dynamics of the interaction between light and matter, the dynamic nonlinear responses of matter, and can selectively control the properties of matter on the fs time scale. The proposed technique can be an extraordinary

tool to study the structural responses of materials to THz radiation in the manner of high space-time resolution.

**Author Contributions:** Conceptualization, K.Z. and C.F.; methodology, Y.K., K.Z. and C.F.; software, Y.K., Z.W. and K.Z.; validation, Y.K., Z.W., K.Z. and C.F.; formal analysis, Y.K., K.Z. and C.F.; investigation, Y.K. and K.Z.; data curation, Y.K.; writing—original draft preparation, Y.K.; writing—review and editing, Y.K., Z.W., K.Z. and C.F.; visualization, Y.K.; supervision, K.Z. and C.F.; project administration, K.Z. and C.F.; funding acquisition, K.Z. and C.F. All authors have read and agreed to the published version of the manuscript.

**Funding:** This research was funded by the National Natural Science Foundation of China, grant number 12105347, 12275340.

**Institutional Review Board Statement:** Not applicable.

**Informed Consent Statement:** Not applicable.

**Data Availability Statement:** Data sharing is not applicable to this article.

**Acknowledgments:** The authors would like to thank Hao Sun, Lingjun Tu, Yaozong Xiao, and Hanxiang Yang for their fruitful discussions on FEL physics and simulations.

**Conflicts of Interest:** The authors declare no conflict of interest.

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
