# Peer review of "Generating High-Power, Frequency Tunable Coherent THz Pulse in an X-ray Free-Electron Laser for THz Pump and X-ray Probe Experiments"

_photonics, doi:10.3390/photonics10020133_

Round 1

Reviewer 1 Report

The authors presented simulation results to produce X-ray FEL and THz pulses with single electron bunch. The method adopted in the manuscript is to utilize frequency beating to produce THz bunching on the beam before sending it to THz wiggler for lasing. And the beam with THz bunching was able to produce FEL pulses first with a shorter undulator period. This scheme could be a good candidate plan for THz-pump FEL-probe scheme in FEL facilities. In this method, THz pulses are generated after FEL pulses, so it requires time delay of X-ray pulses to achieve correct time sequence for pump/probe.

The scheme in the manuscript was good to me at the first read. However, after I checked the previous paper from the same group, Ref. [43] in the manuscript, I have some doubts about the significance of this work. If we compare the two papers, we will find the theoretical parts (frequency beating and beam modulation) are very similar. The main difference is a new set of parameters for different electron beam energies (ten of MeV vs. 1.4GeV) and the inclusion of simulations results for FEL pulse. These differences are not enough to convince me the importance and novelty of this work. So I cannot recommend the manuscript for publication in this journal before the authors could provide more explanations about this.

Reviewer 2 Report

This paper proposes a scheme to generate synchronized high-power X-ray pulses with strong field and frequency tunable coherent THz pulses, via simulations based on realistic beam parameters of Shanghai soft X-ray free-electron laser facility. The paper is sound and makes a good case for a new development.

I would suggest the authors to update their paper when discussing other XFELs (references 38 to 42) by citing the more recent papers, e.g. for swissFEL, see Prat et al, Nature Photonics14 (2020) 748.

Reviewer 3 Report

The authors have proposed an X-ray FEL-based technique using single electron beam to generate X-ray and THz radiation simultaneously with both pulse energies at mJ level, which can be used to THz pump and X-ray probe experiments for the dynamic research on the interaction between THz pulse and matter at femtosecond time scale. It sounds and could be published with few minor modifications,

-Lines 137-139, How by changing |α| and τ to change the THZ wavelength? Please give the calculation formulas or relevant explanation (attach the references).

-In the Figure 4 caption, ‘left’ or ‘right’ labels may cause confusion, it is better to change to ‘red line’ or ‘blue line’ with γ and I in front, respectively.

-In Figures 5 and 7, duplicate labels of (a-d) appear in and below the figures.

Reviewer 4 Report

Precisely synchronized X-ray and strong-field coherent terahertz (THz) enables coherent THz excitation of many fundamental modes (THz pump) and capturing the X-ray dynamic image of matters (X-ray probe), while the generation of such light source is still a challenge for the most existing techniques. In this paper, a novel X-ray free-electron laser based light source is proposed to produce synchronized high-power X-ray pulse and strong field, widely frequency tunable coherent THz pulse simultaneously. The technique adopts a frequency beating laser modulated electron bunch with a Giga-electron-volt beam energy to generate X-ray pulse and THz pulse sequentially by passing two individual undulator sections with different magnetic periods.

The Manuscript should make these changes.

In the Introduction section, mention other works in the THz field to make it interesting for the readers in different fields. My suggestion is the following sources:

1. C. Chaccour, M. N. Soorki, W. Saad, M. Bennis, P. Popovski and M. Debbah, "Seven Defining Features of Terahertz (THz) Wireless Systems: A Fellowship of Communication and Sensing," in IEEE Communications Surveys & Tutorials, vol. 24, no. 2, pp. 967-993, Secondquarter 2022, doi: 10.1109/COMST.2022.3143454.

2. D. Serghiou, M. Khalily, T. W. C. Brown and R. Tafazolli, "Terahertz Channel Propagation Phenomena, Measurement Techniques and Modeling for 6G Wireless Communication Applications: A Survey, Open Challenges and Future Research Directions," in IEEE Communications Surveys & Tutorials, vol. 24, no. 4, pp. 1957-1996, Fourthquarter 2022, doi: 10.1109/COMST.2022.3205505.

3. Beiranvand, B.; Sobolev, A.S.; Larionov, M.Y.; Kuzmin, L.S. A Distributed Terahertz Metasurface with Cold-Electron Bolometers for Cosmology Missions. Appl. Sci. 2021, 11, 4459

4. Behrokh, B.; Sobolev, A.S.; Anton, V.; Kudryashov, A.V. Composite right/left-handed transmission line with array of thermocouples for generating terahertz radiation. Appl. Phys. 2020, 92, 20502.

5. Zhong, Y.-F.; Ren, J.-J.; Li, L.-J.; Zhang, J.-Y.; Zhang, D.-D.; Gu, J.; Xue, J.-W.; Chen, Q. Measurement of Stress Optical Coefficient for Silicone Adhesive Based on Terahertz Time Domain Spectroscopy. Photonics 2022, 9, 929. https://doi.org/10.3390/photonics9120929

The mechanism of Figure 2 is not clear to the reader. Please explain more from a technical point of view.

How and with what method was Figure 4 measured?

The longitudinal phase spaces before and after the dispersion section, describe its advantages and disadvantages.

What is the difference between the results in Figure 3 and Figure 8?

Describe Discussion and Conclusion separately and compare your work with similar works.

Round 2

Reviewer 1 Report

Thanks for the explanations from the authors for my questions. I agree the two papers are focusing on different problems though their theoretical part and method are very similar. The requirement for THz-pump and FEL-probe in FEL facilities would attract readers' interest in this manuscript and the method proposed here would be a good candidate, especially for seeded FEL facilities. I recommend the paper for publication after a general editor checking. 

Reviewer 4 Report

All changes are added. I accept the manuscript to be published as a paper.